# Pathogenicity of Different Betanodavirus RGNNV/SJNNV Reassortant Strains in European Sea Bass

**DOI:** 10.3390/pathogens11040458

**Published:** 2022-04-11

**Authors:** Lorena Biasini, Paola Berto, Miriam Abbadi, Alessandra Buratin, Marica Toson, Andrea Marsella, Anna Toffan, Francesco Pascoli

**Affiliations:** Istituto Zooprofilattico Sperimentale delle Venezie, Legnaro, 35020 Padova, Italy; lbiasini@izsvenezie.it (L.B.); pberto@izsvenezie.it (P.B.); mabbadi@izsvenezie.it (M.A.); aburatin@izsvenezie.it (A.B.); mtoson@izsvenezie.it (M.T.); amarsella@izsvenezie.it (A.M.); atoffan@izsvenezie.it (A.T.)

**Keywords:** betanodavirus, viral encephalopathy and retinopathy, reassortant, European sea bass, pathogenicity

## Abstract

European sea bass (*Dicentrarchus labrax*) is an important farmed marine species for Mediterranean aquaculture. Outbreaks of betanodavirus represent one of the main infectious threats for this species. The red-spotted grouper nervous necrosis virus genotype (RGNNV) is the most widely spread in Southern Europe, while the striped jack nervous necrosis virus genotype (SJNNV) has been rarely detected. The existence of natural reassortants between these genotypes has been demonstrated, the RGNNV/SJNNV strain being the most common. This study aimed to evaluate the pathogenicity of different RGNNV/SJNNV strains in European sea bass. A selection of nine European reassortants together with parental RGNNV and SJNNV strains were used to perform *in vivo* experimental challenges via intramuscular injection. Additional *in vivo* experimental challenges were performed by bath immersion in order to mimic the natural infection route of the virus. Overall, results on survival rates confirmed the susceptibility of European sea bass to reassortants and showed different levels of induced mortalities. Results obtained by RT-qPCR also highlighted high viral loads in asymptomatic survivors, suggesting a possible reservoir role of this species. Our findings on the comparison of complete genomic segments of all reassortants have shed light on different amino acid residues likely involved in the variable pathogenicity of RGNNV/SJNNV strains in European sea bass.

## 1. Introduction

Viral encephalopathy and retinopathy (VER), also known as viral nervous necrosis (VNN), is a severe infective disease characterized by neuropathological changes associated with high mortality in several fish species. Affected fish usually show abnormal swimming behavior, such as erratic swimming in circles, lying on their sides or upside down; some keep a vertical position with their head or the caudal peduncle above the water surface [1]. The etiological agent is a virus belonging to the *Nodaviridae* family, genus *Betanodavirus*, a naked RNA virus characterized by an extremely high resistance to chemical and physical agents [1].

Betanodavirus is one of the most significant viral pathogens of finfish and represents the main bottleneck for mariculture development in several countries, with particular reference to those surrounding the Mediterranean Sea [2,3,4,5]. Nervous necrosis virus (NNV) can infect several different fish species both in marine and in the freshwater environment; betanodavirus is able to evade the host protective systems and can either replicate and transmit progeny to other cells or remain in a latent condition in the nervous tissues [6]. NNV is a small virus with a diameter of approximately 25–30 nm and an icosahedral symmetry. The viral genome is composed of bi-segmented, single-stranded, positive-sense RNA molecules, RNA1 (3.1 Kb) and RNA2 (1.4 Kb), which encode the RNA-dependent RNA polymerase (RdRp) and the capsid protein (CP), respectively [7]. A subgenomic transcript, RNA3 (0.4 kb), originates from the 3′ terminus of RNA1 during viral replication and encodes the B2 non-structural protein, an inhibitor of cell-RNA silencing [8,9].

Betanodaviruses are classified into four different genotypes: striped jack nervous necrosis virus (SJNNV), tiger puffer nervous necrosis virus (TPNNV), barfin flounder nervous necrosis virus (BFNNV) and red-spotted grouper nervous necrosis virus (RGNNV) [10]. Natural reassortment events can occur between the RGNNV and SJNNV genotypes, and two types of reassortants have been identified to date. The RGNNV/SJNNV and SJNNV/RGNNV genotype name designation refers to the parental genotype of the polymerase/capsid protein genes (RNA1/RNA2, respectively) [1,11,12].

Of all the betanodaviruses, the RGNNV genotype has the broadest range of warm-water hosts and has the widest geographic distribution throughout Asia, the USA, Australia and the Mediterranean basin [13]. On the other hand, a study on 120 field isolates from different outbreaks in the Mediterranean Sea conducted by Panzarin et al. [12] highlighted that almost 20% of the circulating betanodavirus species are RGNNV/SJNNV reassortants. Moreover, in the last few years an increasing number of betanodavirus outbreaks caused by RGNNV/SJNNV strains have been observed in Gilthead sea bream (*Spaurs aurata*) farms [14,15,16,17].

European sea bass (*Dicentrarchus labrax*; ESB) is the most farmed marine species in the Mediterranean Sea [18] and is also highly susceptible to VER. In this species, mortality mainly varies depending on fish age and water temperature. Indeed, outbreaks caused by RGNNV genotype (hereafter RG) can be devastating in hatcheries, with mortality reaching 80–100%, and even older fish can be seriously affected, although this occurs less frequently [19,20,21]. The SJNNV genotype (hereafter SJ) has been detected during natural outbreaks of reared marine species, such as *Pseudocaranx dentex* and *Solea senegalensis*. In wild or farmed ESB, no mortality has been recorded in association with the SJ genotype [13]. However, in a few experimental studies, strains belonging to the SJ genotype have been used in challenges of ESB with no or low mortality (0–10%) [22,23,24]. Finally, a natural outbreak caused by reassortant RGNNV/SJNNV (hereafter RG/SJ) strain has been described in ESB larvae and juveniles with a cumulative mortality of 10% [16]. In addition, RG/SJ strains were used in some experimental infections of ESB, causing a cumulative mortality of variable rates (≤ 25%) [22,23,24]. Nevertheless, such mentioned studies have examined only few RG/SJ strains and the mechanisms related to the reduced pathogenicity in this fish species need to be further clarified. To date, only a small number of studies have focused on NNV virulence determinants and identified two amino acids on the C-terminal of the capsid protein, at position 247 and 270, involved in the virulence of betanodaviruses to ESB [25,26].

In the present investigation, we aimed to explore the pathogenesis of nine reassortant RG/SJ strains in European sea bass by comparing the virulence of the different viral strains through two distinct routes of infections and highlighting the susceptibility of this fish species to this particular betanodavirus genotype.

## 2. Results

### 2.1. Clinical Observation and Survivorship Curves

Information regarding all the strains used for the experimental infections carried out in the present study are reported in Table 1. For details on experimental trials, see Section 4.3. ESB showed no significant mortalities during the acclimation period of 10 days before trials. Moreover, virological analyses carried out on both cell culture and with rRT-PCR confirmed the negativity of the ESB batch for the most common pathogens, including betanodavirus; therefore it was considered suitable for the experimental trials.

Fish challenged by intramuscular injection showed typical clinical signs associated with VNN starting from 5 days post infection (dpi), with slight differences in all the infected groups, except for groups SJ_484-2_ and SJ_540-7_ which never showed any symptoms. Diseased fish suffered anorexia and showed abnormal swimming behavior characterized by whirling and surfing movements. The peak of mortality was recorded between 5 and 7 dpi. On the other hand, in the bath immersion challenge, the typical clinical signs associated with VER and massive mortalities occurred later over time, starting at 7 dpi for the RG_283_ group and at 9 dpi for the RG/SJ groups.

Cumulative mortality percentages recorded in the intramuscular-challenged control groups were 34.55% and 1.82% in RG_283_ and SJ_540-7_, respectively, while it was absent in SJ_484-2_ and mock groups. The overall results obtained in this challenge highlighted a great variation in terms of mortality in the groups infected with the reassortant strains, with values ranging between 2.04% and 26%. In particular, the RG/SJ_132_ group showed a significant decrease in survival probability, reaching 26% of mortality. The RG/SJ_292-7.8_ and RG/SJ_292-1.2_ groups exhibited mortality of 15.69% and 12.73%, respectively. The remaining six groups infected with reassortant strains (RG/SJ_165-6_, RG/SJ_187_, RG/SJ_188_, RG/SJ_367-2_, RG/SJ_461-1_, RG/SJ_61-48_) induced low levels of mortality in infected fish. In the bath immersion challenge, the differences in terms of mortality estimates appeared less evident in comparison to the results obtained by intramuscular infection. Indeed, cumulative mortality percentages were lower in all the RG/SJ infected groups (*p*-value < 0.05), ranging between 3.7 and 7.4%, while in the RG_283_-infected group, mortality reached 18.5%. Detailed results on cumulative mortality related to each infected group are reported in Table 2.

For both challenges, Kaplan–Meier curves were also developed by plotting the survival rates during the observation period (Figure 1). Statistical analysis performed on intramuscular challenge data highlighted a high similarity between RG/SJ_132_ and the RG_283_ reference strain (*p*-value > 0.05). Moreover, RG/SJ_292-1.2_ and RG/SJ_292-7.8_ resulted as a homogeneous group (*p*-value > 0.05) inducing a significantly intermediate level of mortality. The remaining RG/SJ groups exhibited low mortalities similarly to SJ reference groups (*p*-value > 0.05). It should be noted that no correlation was found between the cumulative mortality and the viral load in the infected groups (R^2^ = 0.256). For the bath immersion challenge, Kaplan–Meier curves showed a significant difference between the RG_283_ reference group and all the RG/SJ groups.

### 2.2. Molecular Viral Detection in Experimentally Infected Fish

All dead (*n* = 67) and survived (*n* = 60) juvenile sea bass from the intramuscular infection trial yielded positive results by qualitative rRT-PCR for the detection of nervous necrosis virus. In contrast, no positivity was detected in tested survivors (*n* = 5) from the mock-infected group (data not shown). Brain samples from survived juvenile sea bass (*n* = 40 infected, *n* = 5 mock) were also tested using quantitative RT-qPCR in order to estimate the viral loads in apparently healthy fish 28 days after infection. Total RNAs were extracted from sampled brains and exhibited excellent concentrations and integrity. In general, the presence of NNV, in terms of LCN values, was detected in all the samples from the RG/SJ-treated groups with a homogeneous trend among groups, ranging between 4.34 ± 0.31 and 5.01 ± 0.47. Different, slightly higher viral loads were found in samples from the RG_283_ control group (LCN value of 5.26 ± 0.16, *p*-value < 0.05) (Table 2). RT-qPCR results are shown in Figure 2A.

Furthermore, in the bath immersion experimental trial, all samples from dead (*n* = 20) and survivor (*n* = 20) fish tested positive for betanodavirus through qualitative real-time RT-PCR. Conversely, survivors (*n* = 5) from the mock-infected group tested negative for the presence of NNV. Viral loads were estimated in survivors of all the tested groups (*n* = 20 infected, *n* = 5 mock). All samples from the RG/SJ-treated groups exhibited a similar trend in terms of LCN values, ranging from 4.27 ± 0.58 to 4.79 ± 0.37. A LCN value of 4.67 ± 0.39 was detected in the RG_283_ reference group (Table 2). Altogether, results displayed no statistically significant differences in the groups infected by bath immersion. RT-qPCR results are reported in Figure 2B.

### 2.3. Genetic Characterization and Phylogenetic Analysis

The maximum likelihood phylogenetic trees inferred for the RNA1 and RNA2 complete genetic segments confirmed the RGNNV genotype of strain 283 and the SJNNV genotype of strains 484-2 and 540-7. In addition, the phylogenies also identified strains 132, 292-7.8, 292-1.2, 187, 367-2, 61-48, 188, 461-1 and 165-6 as members of the reassortant RGNNV/SJNNV genotype.

Nucleotide similarity calculated among the challenge viruses ranged from 81 to 100% for RNA1 and from 78 to 100% for RNA2, while the amino acid identity ranged from 85 to 100% for RNA1 and from 72 to 100% for RNA2.

Pairwise nucleotide and amino acid distances estimated with the MEGA 7.0 package are provided in the Appendix A, while the phylogenetic trees describing the genetic relationships existing between all 12 viral strains are shown in Figure 3.

### 2.4. Protein-Structure Prediction

Both RNA1 and RNA2 viral segments of all the tested strains used in this work have been completely sequenced and deposited in Genbank (Table 1). In order to investigate any difference that could allow the identification of potential virulence markers, both nucleotide and amino acid alignments were obtained for the employed reassortant and reference strains. RNA1 nucleotide alignment highlighted a great variability at the 3′-end, in correspondence of the subgenomic RNA3 encoding for a 75-aa-long protein, B2. The B2 sequence analysis of the RG/SJ strains with respect to the RG and SJ strains displayed several amino-acid substitutions surrounding a proline-rich region. The main differences observed in comparison to the 283 parental strain were P63L, L64V/A/T, P65L, I68T and E69V. The 3D modeling of B2 (by i-Tasser server, data not shown) placed these residues in a loop at the C-terminal portion of the protein, highlighting the impact of the observed substitutions in changing the hydrophobicity and steric hindrance of the environment. Moreover, we compared the hypothetical structure of VNNV B2 protein with two similar alphanodaviruses, Flock House and Nodamura Virus (PDB: 2B9Z and 3G80, respectively). Results allowed the detection of residues R52, R53, R59 and R60 in all the RG/SJ strains that could probably play an analogous role similar to the one of the arginines reported in alphanodaviruses and involved in the protection of viral replication.

While analyzing the sequence of the capsid protein, encoded by the RNA2 gene, a high variability was observed in the P-domain region (residues 222–338, protrusion domain at C-terminal of the protein) involved in the host–cell-surface interaction and in the trimerization of the protein. In all the tested RG/SJ reassortant strains, we identified the hypothetical pathogenicity markers, S247 and S270, previously described by Moreno et al. [26]. In addition, seven other variable amino-acids were observed within the capsid P-domain region. It is noteworthy to highlight that the seven newly identified residues are all externally exposed to the structure of the P-domain (Figure 4). Interestingly, six reassortants out of nine showed Q292S, V293I or F298Y substitutions (color coded in blue in Figure 4), in comparison to the parental SJ strains. On the other hand, all the RG/SJ strains, with respect to both SJ and RG reference strains, showed substitution A223T (M in RG strain), A291T (A in RG strain) and F297Y (W in RG strain) (color coded in red in Figure 4). Moreover, as regards the amino acids at position 297, the mentioned substitutions could play a role in affecting the hydrophobicity, varying in the SJ (less virulent), RG (more virulent) and reassortant strains. It is also interesting to observe, in position 299, the presence of a positive charge (R) in parental RG strain and a negative one (D) in SJ and reassortant strains (color coded in yellow in Figure 4). All together, this information denotes a hydrophobic region with a weak positive charge in the parental RG strain, while in the SJ parental strains the same region is negatively charged, less hydrophobic and with a smaller steric hindrance. Additionally, analyzed reassortants exhibit an intermediate feature, with a hydrophobic surface presenting a weak negative charge. Finally, it is worth mentioning that the P-domain region between residues 291 and 302 is arranged as a flexible external loop and should be involved in host specificity and/or in changing viral affinity for the neural receptors.

## 3. Discussion

To date, reported *in vivo* experimental trials have compared very few RGNNV/SJNNV reassortants to understand the pathogenicity of these strains in European sea bass. The described results confirmed the pathogenicity of this genotype in ESB juveniles, inducing different levels of cumulative mortalities [22,23,24]. However, *in vivo* pathogenicity mechanisms and molecular determinants of virulence have not been fully described.

In our study, we aimed to investigate the pathogenesis of a considerable number of reassortant RG/SJ strains, together with their parental reference strains, by performing two different *in vivo* experimental challenges in ESB.

The observed mortalities were in agreement with data reported by Vendramin et al. [22] and highlighted a variable susceptibility of this fish species to all utilized reassortant strains. Indeed, mortality induced by reassortant isolates ranged from 2.04 to 26% in the intramuscular injection trial and from 3.7 to 7.4% in the bath immersion trial. It is worth mentioning that, since the pathogenicity of RG/SJ and SJ strains is usually reported as very low in ESB [16,22,23,24], intramuscular infections with the RG/SJ and SJ strains were performed using the maximal available viral doses in order to simulate the worst scenario and maximize the effects of the viruses. On the other hand, the RG strain was 2 Log-fold diluted prior to use in order to avoid 100% mortalities.

Despite the variability of the viral titers, the analysis of Kaplan–Meier curves in the intramuscular injection trial allowed us to distinguish three homogeneous groups. Indeed, it was possible to classify reassortant strains according to their virulence phenotypes: (i) high for strain 132, (ii) intermediate for strains 292.1-2 and 292.7-8 and (iii) low for all the other RG/SJ strains. Nonetheless, within the identified groups, the observed mortality induced by each strain did not appear directly correlated to the viral titer used. On the other hand, the viral titer in the bath immersion challenge was normalized to better compare the results obtained from each group. Although the distinction into three classes of virulence was not confirmed in the bath infection trial, overall results showed that reassortant strains induced homogeneous lower mortality in ESB in comparison to parental RG reference strain. However, RT-qPCR results displayed similar LCN values in the brain of the survivor fish of the RG/SJ infection groups, confirming the ability of reassortant strains to enter through the natural route, infect and be retained in the nervous tissues of ESB.

While investigating the presence of the virus through rRT-PCR in the target organ (brain) of all the survivor fish, in both challenges, it was interesting to observe that they all tested positive for NNV, thus suggesting that ESB could play an important role as carriers of betanodavirus to more susceptible species sharing the same environment (e.g., Gilthead sea bream and sole). Indeed, horizontal transmission of betanodavirus among farmed species has been previously reported by several authors [16,27,28]. Moreover, our pathogenicity and sequencing analysis further supported the hypothesis of horizontal transmission of the virus in farmed fish. Analysis always paired together strains isolated from the same farm and year, but from different species, and this may explain the common origin of the viruses. More precisely, the paired strains isolated from ESB and sea bream of the same farm were 292-1.2 with 292-7.8, and 187 with 188. In addition, confirmation that the experimentally infected survivors act as carriers of betanodavirus is given by the results obtained through the molecular viral investigation performed by RT-qPCR. Overall, data of both challenges showed high and persistent viral loads, in terms of LCN values, in all the tested brains after 28 days post infection.

Additionally, to investigate the presence of potential virulence markers in reassortant strains, both nucleotide and amino acid alignments were obtained and scrutinized. It is worth mentioning that the viral factors contributing to the pathogenicity and virulence of betanodavirus have not yet been clearly characterized. Interestingly, it has been reported that RNA2 gene encoding for the capsid protein plays an important role in viral host specificity and, more precisely, in the identification of good candidates for host specificity in the protruding domains of the C-terminal region [29,30]. More recently, several studies have indicated the presence of the two residues in the protruding domain playing a major role in betanodavirus pathogenicity [11,25]. In particular, aa 247 and 270 have been identified as putative receptor binding sites; hence, substitutions at these positions could affect virus–host interaction [25,26]. Our findings have partially confirmed previously reported results as all reassortant strains considered in this work bear serine residues in the aa positions 247 and 270. Additionally, obtained outcomes also suggested the possible existence of additional interesting residues, all exposed externally to the structure of the trimeric P-domains. In particular, seven variable amino acids were identified (aa positions 223, 291–293, 297–299) and through a protein-structure prediction it was possible to speculate that modifications in these positions could have a consequence in changing the hydrophobicity and the steric hindrance of the exposed surface region, resulting in the alteration of the physiochemical properties of the capsid protein.

Additionally, while considering the subgenomic RNA3 gene (protein B2) in the analysis, RG/SJ strains displayed several amino acid substitutions in a proline-rich region, within a loop at the C-terminal portion of the protein, which could alter the interactions with molecular targets. Additionally, four conserved arginine residues were also detected in all the reassortant strains that are probably involved in protecting viral replication. Interestingly, in other studies it has been reported that in two alphanodaviruses, namely Flock House and Nodamura Virus, the B2 protein is able to bind dsRNAs with some residues, including arginines, hence negatively impacting the host cellular antiviral machinery by suppressing the RNAi and allowing viral RNA accumulation [31,32]. Finally, in the current study, we also observed differences in the non-coding region (NCR) of both genomic segments with respect to the parental SJ and RG genotypes (data not shown). It is known that NCR may play an important role in virulence, presumably through its effects on genome translation, replication or transcription [33,34]. Nevertheless, more extensive studies are necessary to fully understand and clarify the significance of the observed nucleotidic and amino acidic changes in the reassortant strains, aiming to find a link with aspects such as affinity for cell receptors, host tropism and viral replication kinetics.

## 4. Materials and Methods

### 4.1. Fish

A batch of 1022 ESB juveniles of 5 ± 1 g were purchased from an Italian commercial farm with no recent history of VNN disease. Upon arrival in the experimental aquarium at the Istituto Zooprofilattico Sperimentale delle Venezie (IZSVe), 50 fish were checked for the most common pathogens (parasites, bacteria and betanodavirus). The fish were equally distributed into independent 250 L closed-system tanks equipped with a biological filter, heater/refrigerator and controlled photoperiod. Artificial saltwater was made from carbon filtered and UV exposed tap water added with balanced salt mix (Instant Ocean—Spectrum Brands, Blacksburg, VA, USA), and the salinity was set at 25 Practical Salinity Unit (PSU). Fish were fed daily with a commercial diet (Veronesi Vita 2). The water temperature was maintained at 21 ± 1 °C, oxygen at 7 ± 0.5 ppm and the artificial photoperiod applied was 10 h of light and 14 h of dark.

### 4.2. Virus Isolation and Propagation

Nine RG/SJ reassortant betanodavirus strains isolated in past years (2005–2018) from natural outbreaks in Europe were selected for the present study. Strain selection was made in order to obtain the highest variability in terms of country, year of isolation and species of origin. Three non-reassortant strains were selected as control groups: one virus strain with the RG genotype was selected as a positive reference control, known for its high pathogenicity in ESB (283), and two virus strains of the SJ genotype known to be low pathogenic in ESB (484-2 and 540-7).

Strains were replicated on E-11 cells (a clone of cell line SSN-1) [35] according to the standard procedure [36,37]. The viral titer was calculated by titration on striped snakehead fish cells (SSN-1) [38] according to the Reed and Muench formula [39] and expressed as TCID_50_/mL.

Information regarding the viruses used for the experimental infections carried out in the present study is reported in Table 1.

### 4.3. Experimental Challenges

#### 4.3.1. Challenge by Intramuscular Injection

Juvenile ESB were divided into thirteen experimental groups of 54 fish. Upon sedation (Tricaine-Pharmaq^®^, Overhalla, Norway), fish were intramuscularly injected with 0.1 mL of viral inoculum (TCID_50_/mL ranged between 10^5.80^ and 10^7.55^; see Table 2). For each strain, the inoculum was prepared from infected cell culture supernatant used at the maximal available viral titer. As an exception, for the RG (283) strain, a highly pathogenic genotype for ESB that causes elevated mortalities, the injection concentration was diluted.

Exhaustive information on all the betanodavirus strains selected for this experiment is reported in Table 1 and Table 2. In detail, nine groups were injected with different reassortant RG/SJ betanodavirus strains. Additionally, three comparative reference groups were used and divided as follows: one group was infected with a parental RG genotype strain (283), known to be highly pathogenic in ESB, and two groups were infected with different parental SJ genotype strains (484-2 and 540-7) expected to have low pathogenesis in ESB. One group out of thirteen was injected with the same amount of sterile PBS and used as a mock-infected group.

Fish were regularly monitored over a period of 28 days. Daily mortality was recorded, and moribund and dead fish were collected. At the end of the observation period, survivor fish were euthanized by overdose of anesthetic (Tricaine-Pharmaq^®^, Overhalla, Norway). Five survivor fish from each group were randomly selected and their brains were collected individually and stored in RNA*later*^®^ (Invitrogen by ThermoFisher Scientific, Waltham, MA, USA) for subsequent molecular investigations.

#### 4.3.2. Challenge by Bath Immersion

This second *in vivo* experiment was performed in the form of bath immersion with selected viruses in order to mimic the natural process of infection, and was initiated after the intramuscular challenges.

Five experimental groups, each consisting of 54 juvenile ESB, were used. In detail, three groups were infected with the RG/SJ reassortant strains (132, 292-7.8 and 461-1), selected based on the results obtained from the intramuscular injection trial and with different degrees of pathogenicity. One group was infected with the parental reference strain 283. The last group was mock infected with sterile PBS.

The challenge was performed by immersion for 3 h in static water, reducing the level from 250 to 20 L. The appropriate amount of virus was poured directly into the tank to reach final values of theoretical 10^5^ TCID_50_/mL, which was the maximum concentration achievable by dilution to obtain the same dose for each virus. Additional aeration was supplied during the infection. The final infectious titer was verified by back titration of the water. At the end of the challenge, water level was restored by adding clean saltwater. In this experiment, the observation period lasted 28 days and 5 brains from randomly selected survivor fish from each group were collected and stored in RNA*later*^®^ (Invitrogen by ThermoFisher Scientific, Waltham, MA, USA) for further molecular analysis.

### 4.4. Molecular Analysis for Viral RNA Detection in Experimentally Infected European Sea Bass

#### 4.4.1. Qualitative Real-Time RT-PCR (rRT-PCR)

Qualitative real-time RT-PCR for NNV detection was performed on samples from both intramuscular and bath immersion challenges. Brain samples from all dead and symptomatic fish were singularly tested. In addition, brains from 5 survivor fish from each group (reassortant and control groups) were also tested. Briefly, brain samples were manually homogenized in vials containing sterile quartz sand and E-MEM (Sigma-Aldrich, St. Louis, MO, USA) in a 1:3 *w*/*v* ratio. Homogenates were subsequently clarified by centrifugation at 8000× *g* for 2 min and supernatants collected.

Total nucleic acids were purified from 250 μL of supernatant of each sample using QIAsymphony DSP Virus/Pathogen Midi kit (Qiagen, Hilden, Germany), in combination with the automated system QIAsymphony SP (Qiagen, Hilden, Germany). Isolation of the nucleic acids was performed following the manufacturer’s recommendations. The detection of betanodavirus was performed by applying rRT-PCR targeting RNA1 according to Baud et al. [40].

#### 4.4.2. Quantification of RNA1 Genome Copy Numbers in Experimentally Infected Fish

Brain samples collected from five survivor fish of each infected group were tested by a two-step quantitative real-time PCR (RT-qPCR), targeting viral segment RNA1, as previously described by Toffan et al. [41]. Samples infected with 540-7 and 484-2 strains were not analyzed because the primers were not suitable for the SJ genotype. In addition, intramuscularly infected samples from groups RG/SJ_292-7.8_ and RG/SJ_188_ were not tested as the strains used presented a high nucleotide identity (≥99.9%) with strains 292-1.2 and 187, respectively. Briefly, total RNA from each sample was isolated using RNeasy Mini kit (Qiagen, Hilden, Germany) according to the manufacturer′s “Animal Tissues” purification protocol. RNA integrity of the samples was assessed with Agilent 2100 Bioanalyzer System and RNA 6000 Nano Kit (Agilent Technologies, Santa Clara, CA, USA). In order to preserve RNA long-term integrity, 40 units of RNasin^®^ Plus RNase Inhibitor (Promega Corporation, Fitchburg, WI, USA) were added to each sample. RNA concentration was measured using the Qubit™ RNA BR Assay Kit with the Qubit™ 4 Fluorometer (ThermoFisher Scientific, Waltham, MA, USA) and all RNA samples were normalized at the same concentration in molecular-grade water.

The SuperScript™ III Reverse Transcriptase kit (Invitrogen by ThermoFisher Scientific Waltham, MA, USA) was used for the synthesis of complementary DNA (cDNA). In brief, cDNA was synthesized starting from 0.8 µg of total RNA, 50 pmol of random hexamers and 10 mmol of dNTP. A pre-incubation phase was performed at 65 °C for 5 min followed by 5 min on ice. The reverse transcription reaction was carried out on 20 μL of final volume at 25 °C for 5 min, 50 °C for 60 min and 70 °C for 15 min.

Real-time qPCR reaction was performed in 25 μL of total volume using the SsoFast™ EvaGreen^®^ Supermix (BioRad, Hercules, CA, USA), 0.5 μM of each primer (RNA1_FOR 5′-ATCACTGACGACTCCGTTCACTACCG-3′, RNA1_REV 5′-CATACATGGTATCCTGGTTGTAGTTCC-3′), and 5 μL of 1:10 diluted cDNA template. The quantitative PCR reaction was performed using a CFX96™ Real Time System (BioRad, Hercules, CA, USA) and the amplification thermal profile consisted of a 30 s activation step at 95 °C followed by 40 cycles of 5 s denaturation at 95 °C, 5 s annealing and extension at 60 °C. The reaction was completed with a temperature melting curve analysis performed from 65 to 95 °C (5 s/step, ramp rate 0.5 °C/s).

Absolute quantification of RNA1 genome viral copy numbers was performed by interpolation of the quantification cycle (Cq) values obtained with the standard curves developed with 5-fold dilution of *in vitro* transcribed RNA tested in triplicate. Synthetic RNA1 was produced as described in Toffan et al. [41], and dilutions ranged from 10^8^ to 10^2^ copy numbers/reaction. For each run, linearity (R^2^), reaction efficiency (E) and slope were extrapolated from the standard curves. The limit of detection (LoD) and limit of quantification (LoQ) determined in this assay corresponded to 10^2^ copies/reaction. The results of viral quantification are expressed as logarithmic values of RNA1 gene copy numbers (LCN) detected in 20 ng of brain total RNA.

### 4.5. Whole Genome Sequencing

Viral strains 61-48, 187, 188 and 540-7 underwent complete genome sequencing. Additionally, previously published partial sequences of strains 132, 165-6, 292-1.2 and 292-7.8 were extended. Total viral RNA was isolated from 100 µL of cell-culture supernatant using RNeasy Mini kit (Qiagen, Hilden, Germany) with some minor modifications to the manufacturer’s instructions. Reverse transcription was performed as described above. Subsequently, each viral cDNA was used as a template in 9 different PCR reactions using the Platinum™ Taq DNA Polymerase kit (Invitrogen by ThermoFisher Scientific, Waltham, MA, USA) and the following cycling conditions were applied: 94 °C for 2 min and 40 cycles of 30 s denaturation at 94 °C; 1 min annealing at 50 °C and 70 s elongation at 72 °C; the reaction was terminated with 5 min elongation at 72 °C. The specific primer sets used for the amplification of complete genomes were provided upon request, as reported in Panzarin et al. [42]. PCR products were analyzed for purity and size by capillary electrophoresis on a QIAxcel Advanced System with QIAxcel DNA High Resolution kit (Qiagen, Hilden, Germany). Amplicons were therefore purified with ExoSAP-IT Express (Applied Biosystems by Thermo Fisher Scientific, Baltics, UAB, Lithuania) and sequenced in both directions using the BrilliantDye™ Terminator (v3.1) Cycle Sequencing kit (NimaGen, Nijmegen, The Netherlands). The products of the sequencing reactions were cleaned up using the BigDye XTerminator™ Purification Kit (Applied Biosystems by Thermo Fisher Scientific, Bedford, MA, USA) and analyzed on a 16-capillary ABI PRISM 3130xl Genetic Analyzer (Applied Biosystems, Foster City, CA, USA).

### 4.6. Phylogenetic Analysis

Sequencing data were assembled and edited with the SeqScape software v3.0 (Applied Biosystems). RNA1 and RNA2 complete sequences related to the challenge viruses were aligned and compared to reference nucleotide sequences available in GenBank using the MEGA 7.0 package [43]. To genetically characterize the viral strains used in the present study, phylogenetic trees based on both genetic segments were inferred using the maximum likelihood (ML) method available in the IQ-Tree software v1.6.9 [44]. The best fitting model of nucleotide substitution was determined with ModelFinder [45]. One thousand bootstrap replicates were performed to assess the robustness of individual nodes of the phylogeny, and only values ≥70% were considered significant. Phylogenetic trees were visualized with the FigTree v1.4 software (http://tree.bio.ed.ac.uk/software/figtree/, accessed on 9 February 2022).

### 4.7. Protein-Structure Prediction

The three-dimensional structure of the parental SJ (540-7) capsid protein was generated by the SWISS-MODEL web server (http://swissmodel.expasy.org, accessed on 7 February 2022) [46] using the crystal structure of grouper nervous necrosis virus-like particle (PDB: 4wiz, template with the highest score), an OSGNNV coat protein (RGNNV genotype, GenBank accession number KT071606). The 3D structure was analyzed with the PyMOL software (v2.0 Scrödinger, LLC) and to evaluate the structural properties of amino-acid changes, the location of the respective sites was mapped to the predicted 3D structure of the SJ strain. The 3D structure of B2 was generated by the integrated platform I-TASSER (https://zhanggroup.org//I-TASSER/, accessed on 7 February 2022) [47] using the sequence of the 283 RG strain.

### 4.8. Statistical Analyses

The Kaplan–Meier method was used to estimate the survival function from lifetime data, which allowed us to draw the survival curve for each group and to measure the length of time the fish would survive the infection. The Kaplan–Meier graph plots the cumulative probability of the surviving fish at each day post infection [48]. To compare the different survival curves, the non-parametric Wilcoxon–Breslow–Gehan test was used for equality of survivor functions.

A chi square test was used to compare cumulative mortality at the end of the study period between independent groups. For a two-paired comparison between the mortality in two viruses, two-sample test of proportions was performed. Kruskal–Wallis ANOVA was applied to test the differences of LCN values among groups. All LCN data were expressed as mean ± standard deviation (SD). The 95% confidence interval (95% CI) was calculated for each proportion through Binomial Exact approximation. The correlation between viral load and cumulative mortality was verified by calculation of the coefficient of determination (R^2^). All statistical analyses were performed using the STATA 12.1 (Stata Corp LLC, College Station, TX, USA) software. Reported graphs were generated and edited using GraphPad Prism Software (v9.0.0 GraphPad Software, San Diego, CA, USA).

## 5. Conclusions

Obtained data confirmed the susceptibility of European sea bass to a reassortant RGNNV/SJNNV genotype with an overall moderate to low virulence phenotype. The results also highlighted considerable reassortant viral amounts in the brain of all the survivor fish, both in the intramuscular infection challenge and in the bath immersion challenge, that mimicked the natural infection route. Such evidence suggests a reservoir role of ESB towards other farmed species. In addition, new variable amino-acid residues have been identified and are probably correlated to the observed intermediate virulence phenotype of reassortant strains. However, further analyses should be performed in order to better investigate the correlation between candidate determinants of virulence and pathogenicity of reassortant strains.

## Figures and Tables

**Figure 1 pathogens-11-00458-f001:**
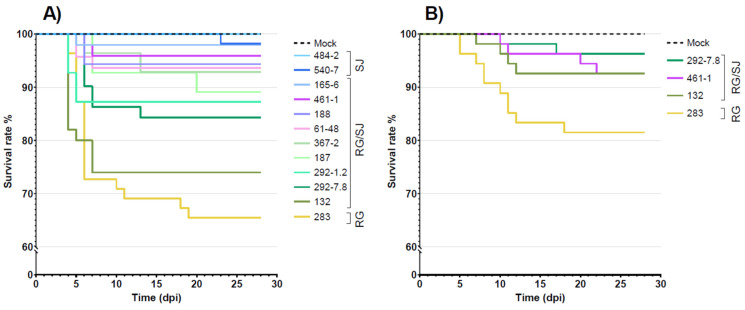
Kaplan–Meier survival curves of the 12 isolates tested for their virulence in ESB. (**A**) Trial by intramuscular injection; (**B**) trial by bath immersion. The *y*-axis reports the survival rate; the *x*-axis reports the observation period expressed as days post infection (dpi). Step curves represent the survival rate of challenged fish in each experimental group.

**Figure 2 pathogens-11-00458-f002:**
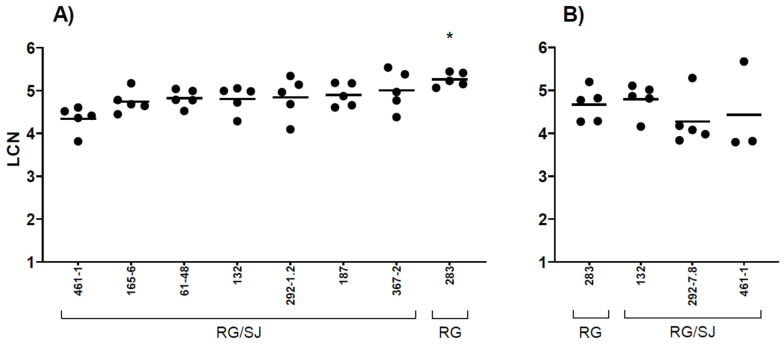
RT-qPCR results in survivors. (**A**) Trial by intramuscular injection; (**B**) trial by bath immersion. The *x*-axis reports the different challenged groups. The *y*-axis reports viral RNA1 gene copy numbers detected in brains. Values are expressed as Log_10_ of copy numbers (LCN) detected in 20 ng of total RNA. For each subject, LCN values are depicted as circles, while the horizontal lines indicate the mean LCN value. Only LCN ≥ LoQ are considered. * = *p*-value < 0.05.

**Figure 3 pathogens-11-00458-f003:**
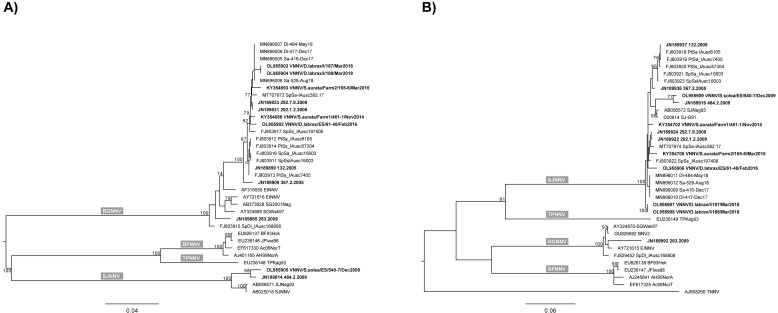
ML phylogenetic trees of (**A**) RNA1 complete ORF and (**B**) RNA2 complete ORF. Isolates used for experimental infections are in bold and compared to betanodavirus representative sequences retrieved from GenBank. The numbers at branch points correspond to bootstrap values expressed as percentages (only values ≥70 are reported). The genotype subdivision according to Nishizawa et al. [10] is shown at the main branches. Scale bars represent nucleotide substitutions per site.

**Figure 4 pathogens-11-00458-f004:**
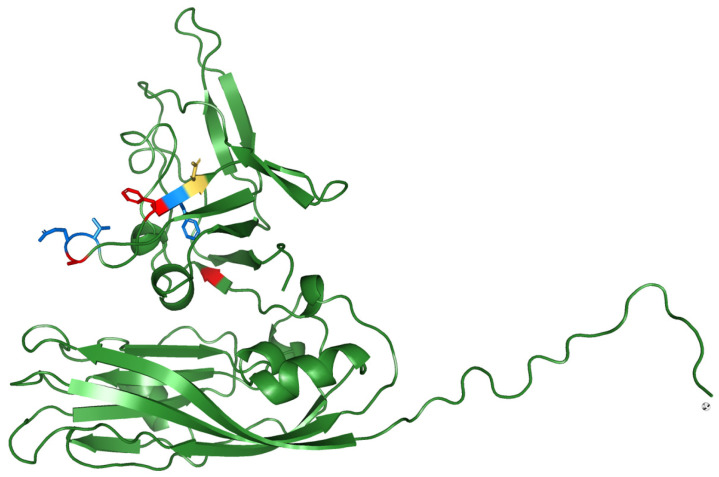
Modeling of reference SJ strain (540-7) capsid protein. In the ribbon, representations of the monomer different colors are reported, indicating the variable amino acids in the P-domain region. Residues color codes are **blue** for 292, 293 and 298; **red** for 223, 291 and 297; **yellow** for 299.

**Table 1 pathogens-11-00458-t001:** Data on fish betanodavirus isolates used in this study for experimental challenges.

Strain	Abbreviation	Year	Host	Origin	Genotype	RNA1	RNA2	Reference
						GenBank Accession No.		
283.2009	283	2009	*D. labrax*	Italy	RG	JN189865.2	JN189992.2	[12]
132.2005	132	2005	*D. labrax*	Italy	RG/SJ	JN189899.2	JN189937.2	[12]
292.7.8.2009	292-7.8	2009	*D. labrax*	Greece	RG/SJ	JN189833.2	JN189924.2	[12]
292.1.2.2009	292-1.2	2009	*S. aurata*	Greece	RG/SJ	JN189831.2	JN189922.2	[12]
VNNV/D.labrax/I/187/Mar2018	187	2018	*D. labrax*	Italy	RG/SJ	OL955903	OL955907	This study
367.2.2005	367-2	2005	*D. labrax*	Italy	RG/SJ	JN189909.2	JN189936.2	[12]
VNNV/D.labrax/ES/61-48/Feb2016	61-48	2016	*D. labrax*	Spain	RG/SJ	OL955902	OL955906	This study
VNNV/S.aurata/I/188/Mar2018	188	2018	*S. aurata*	Italy	RG/SJ	OL955904	OL955908	This study
VNNV/S.aurata/Farm1/461-1/Nov2014	461-1	2014	*S. aurata*	*n.d.*	RG/SJ	KY354688.2	KY354702.2	[15]
VNNV/S.aurata/Farm2/165-6/Mar2016	165-6	2016	*S. aurata*	*n.d.*	RG/SJ	KY354693.2	KY354706.2	[15]
VNNV/S.solea/ES/540-7/Dec2009	540-7	2009	*S. solea*	Spain	SJ	OL955905	OL955909	This study
484.2.2009	484-2	2009	*S. senegalensis*	Spain	SJ	JN189814.2	JN189919.2	[12]

*n.d.* = not disclosed.

**Table 2 pathogens-11-00458-t002:** Results obtained in both intramuscular injection and bath immersion challenges.

Route of Viral Infection	Challenge Group	TCID_50_/mL ^1^	% Mortality (95% CI)	LCN ± SD ^2^
**Intramuscular injection**	RG_283_	10^5.80^	34.55 (22.24–48.58)	5.26 ± 0.16
	RG/SJ_132_	10^6.80^	26.00 (14.63–40.35)	4.81 ± 0.32
	RG/SJ_292-7.8_	10^5.80^	15.69 (7.02–28.59)	*n.e.*
	RG/SJ_292-1.2_	10^5.80^	12.73 (5.27–24.48)	4.84 ± 0.48
	RG/SJ_187_	10^6.05^	10.91 (4.11–22.25)	4.90 ± 0.27
	RG/SJ_367-2_	10^5.80^	7.14 (1.98–17.29)	5.01 ± 0.47
	RG/SJ_61-48_	10^7.05^	6.38 (1.34–17.54)	4.82 ± 0.21
	RG/SJ_188_	10^6.80^	5.66 (1.18–15.66)	*n.e.*
	RG/SJ_461-1_	10^7.55^	4.08 (0.50–13.98)	4.34 ± 0.31
	RG/SJ_165-6_	10^6.55^	2.04 (0.05–10.85)	4.74 ± 0.27
	SJ_540-7_	10^6.80^	1.82 (0.05–9.72)	*n.a.*
	SJ_484-2_	10^7.30^	0.00 (0.00–6.49)	*n.a.*
**Bath immersion**	RG_283_	10^5.05^	18.52 (9.25–31.43)	4.67 ± 0.39
	RG/SJ_132_	10^5.05^	7.41 (2.06–17.89)	4.79 ± 0.37
	RG/SJ_292-7.8_	10^5.05^	3.70 (0.45–12.75)	4.27 ± 0.58
	RG/SJ_461-1_	10^5.05^	7.41 (2.06–17.89)	4.43 ± 1.08 ^#^

^1^ TCID_50_ refers to the mL of inoculum injected in the intramuscular challenge or to the mL of infected water used in the bath immersion challenge; ^2^ LCN is reported as mean logarithmic value of RNA1 gene copy number detected in 20 ng of total RNA of analyzed survivor fish (*n* = 5) and relative standard deviation (SD); ^#^ calculated on 3 fish with values > LoQ; *n.a.* = not analyzed (not suitable primers for strain detection); *n.e.* = not executed (nucleotides identity between strains 187 and 188 and between 292-1.2 and 292-7.8 ≥ 99.9%).

## Data Availability

Nucleotide sequences corresponding to full-length RNA1 and RNA2 generated in the present study have been deposited in GenBank under the following accession numbers: JN189865.2; JN189899.2; JN189833.2; JN189831.2 JN189909.2; JN189814.2; KY354688.2; KY354693.2; OL955902-OL955905 for RNA1 and JN189992.2; JN189937.2; JN189924.2; JN189922.2; JN189936.2; JN189919.2; OL955906-OL955909; KY354702.2; KY354706.2 for RNA2.

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
