# Peer review of "Pathogenicity of Different Betanodavirus RGNNV/SJNNV Reassortant Strains in European Sea Bass"

_pathogens, 2022, doi:10.3390/pathogens11040458_

Round 1
Reviewer 1 Report
I was honored to review the manuscript entitled “Pathogenicity of Different Betanodavirus RGNNV/SJNNV Reassortant Strains in European Sea Bass” submitted to Pathogens Journal. Mediterranean marine aquaculture has suffered significant economic loses due to viral nervous necrosis (VNN) outbreak mainly caused by different RGNNV betanodavirus strains. The reassortant strains caused the massive mortality in European sea bass. This research article could help to open the wide insight about pathogenicity of the different strains and help to cure the disease as well as decrease economic loses. I would like to thank authors for preparing this article in the perfect way. The study presents high quality and has interesting topic however, it needs some correction for improving the quality and needs some English editing. Here are some grammatical and scientific points that I suggested:
- Please add some data about clinical signs in introduction section.
- Line 67: The rate of mortality in case of SJNNV genotype?
- Rewrite line 105-107.
- It would be better to change this work to this study in table 1.
- Please clarify why the authors did not use the same dose during the IM challenge?
- Line 172: 81 to 100% and 78 to 100%
- Line 234: It would be better to change Vandramin and colleague to Vandramin et al. Please change in whole manuscript.
- Please rewrite line 233-239.
- Line 243-244: rewrite because of grammar.
- Line 398-401:please rewrite.
Reviewer 2 Report
In this article, the pathogenesis of nine reassortant RGNNV/SJNNV strains in European sea bass was evaluated by comparing the virulence of the different viral strains through two distinct routes of infections and highlighting the susceptibility of this fish species to this particular betanodavirus genotype. The results are interesting. However, some of point to be considered by the authors are indicated below.
- Abstract: the full name of “RGNNV” and “SJNNV” should be given at their first appearance here.
- In the injection group, 0.1 mL of viral inoculum (TCID50/mL ranged between 10^5.80 and 10^7.55) were used, while in the immersion group, the final values of theoretical were 105 TCID50/mL. please provide the references.
- As shown in Fig. 2B, why just there are three circles representing LCN values in RG461.1 of the bath immersion group?
- 2.1 Clinical observation and survivorship curves
This section is a bit confusing. It is recommended that some paragraphs can be combined into one. Such as, the first and second paragraph; the third paragraph and the describes “Additionally, typical clinical signs ……and at 9 dpi for the RG/SJ groups” in the seventh paragraph are combined into a new one; the fourth and seventh are combined to show the results about the mortality in Table 2 of the injection and immersion groups; The fifth and eighth are combined for explanation of Figure 2.
- As shown in Table 2, the mortality of RG283, 2RG/SJ132, RG/SJ292-7.8 of the immersion group were all lower than that of the injection group, why the mortality of RG/SJ461-1 in the immersion group was higher than that of the injection group? It should be explained.
Reviewer 3 Report
The article "Pathogenicity of different Betanodavirus RGNNV/SJNNV reassortant strains in European Sea Bass" describe susceptibility of European sea bass to reasortant RGNNV/SJNNV genotype and present different virulence of nine reassortant RG/SJ. An article is original, clear, relevant for the field but not presented in a well-structured manner. After Inroduction part should be: 2. Materials and Methods, 3. Results, 4. Discussion and 5. Conclusion. Remember about changing references in the text.
I have also some minor adjustments:
62 - please provide Latin name of ESB
195, 196, 201, 208, 212 - please explain abbreviations
329 - list the names of the strains
330-provide informationin the text which countries they come from
336, 337 - please explain abbreviations of cell lines
340 - in this place should be Table 1.
345 - mark in this place, that all informations are in Table 2
352-356 - list the names of the strains
419-420 - check the font
Round 2
Reviewer 1 Report
I would like to thank the authors for providing this version of the manuscript.